# Optimisation of the Chicken Chorioallantoic Membrane Assay in Uveal Melanoma Research

**DOI:** 10.3390/pharmaceutics14010013

**Published:** 2021-12-22

**Authors:** Ekaterina A. Sokolenko, Utta Berchner-Pfannschmidt, Saskia C. Ting, Kurt W. Schmid, Nikolaos E. Bechrakis, Berthold Seitz, Theodora Tsimpaki, Miriam Monika Kraemer, Miltiadis Fiorentzis

**Affiliations:** 1Department of Ophthalmology, University Hospital Essen, University of Duisburg-Essen, Hufeland Str. 55, 45147 Essen, Germany; utta.berchner-pfannschmidt@uk-essen.de (U.B.-P.); nicolaos.bechrakis@uk-essen.de (N.E.B.); theodora.tsimpaki@gmail.com (T.T.); miriam.kraemer@uk-essen.de (M.M.K.); miltiadis.fiorentzis@uk-essen.de (M.F.); 2Institute of Pathology, University Hospital Essen, University of Duisburg-Essen, Hufeland Str. 55, 45147 Essen, Germany; saskia.ting@uk-essen.de (S.C.T.); kw.schmid@uk-essen.de (K.W.S.); 3Department of Ophthalmology, Saarland University Medical Center, Kirrberger Str. 100, 66421 Homburg, Germany; berthold.seitz@uks.eu

**Keywords:** chicken chorioallantoic membrane assay, chicken embryo, CAM assay, uveal melanoma, metastatic uveal melanoma, in vivo model

## Abstract

The treatment of uveal melanoma and its metastases has not evolved sufficiently over the last decades in comparison to other tumour entities, posing a great challenge in the field of ocular oncology. Despite improvements in the conventional treatment regime and new discoveries about the genetic and molecular background of the primary tumour, effective treatment strategies to either prevent tumours or treat patients with advanced or metastatic disease are still lacking. New therapeutic options are necessary in order to achieve satisfactory local tumour control, reduce the risk of metastasis development, and preserve the eyeball and possibly the visual function of the eye. The development of in vivo model systems remains crucial for the identification and investigation of potential novel treatment modalities. The aim of this study was the optimisation of the chorioallantoic membrane (CAM) model for uveal melanoma research. We analysed the established CAM assay and its modification after the implantation of three-dimensional spheroids. The chorioallantoic membrane of a chick embryo was used to implant uveal melanoma-cell-line-derived spheroids in order to study their growth rate, angiogenic potential, and metastatic capability. Using the UM 92.1, UPMD2, UPMM3, and Mel270 cell lines, we were able to improve the viability of the embryos from 20% to >80% and to achieve up to a fourfold volume increase of the transplanted spheroid masses. The results point to the value of an optimised chicken embryo assay as an in vivo model for testing novel therapies for uveal melanoma by simplifying the research conditions and by contributing to a considerable reduction in animal experiments.

## 1. Introduction

Uveal melanoma (UM) belong to the group of most common primary intraocular malignant tumours in adults. Six to seven new cases per 1 million residents in Europe are diagnosed each year, and it, therefore, remains a rare tumour disease [1]. UM occurs significantly more often in people of White descent than in other ethnic groups. While the disease has no gender preference, the average age of onset is between 60–70 years. In children, UM is extremely rare, but descriptions of individual cases can be found in the literature [2,3,4]. The choroid (90%), ciliary body (7%), and iris (2%) are the most commonly affected structures [5]. The precise aetiology of UM is still unknown. Even though the literature describes familial forms (e.g., BRCA1-associated protein-1 (BAP1) mutation, oculodermal melanosis, familial atypical multiple mole melanoma (FAMMM) syndrome, and neurofibromatosis (NF) type I) as well as phenotypical associations, a hereditary genesis has not yet be confirmed [6,7,8]. In comparison to cutaneous melanoma, mutations such as BRAF, NRAS, and NF1 are uncommon [9].

The molecular pathogenesis of UM is characterised by specific chromosome alterations and gene mutations. The most common abnormalities include loss of the 1p, 3, 6q, and 8p chromosomes as well as a gain of the 1q, 6p, and 8q chromosomes. Inactivating mutations in BAP1, a tumour suppressor gene located on chromosome 3p, are found in approximately 47% of primary UM and 84% of metastatic UM cases, consistent with the association between BAP1 mutations and poor prognosis [10]. Moreover, UM is characterised in approximately 80% of cases by mutually exclusive initiating mutations in the guanine nucleotide-binding protein G(q) subunit alpha (GNAQ), and the quinine nucleotide-binding protein subunit alpha-11 (GNA11), as well as other genes such as CYSLTR2, PLCB4, BAP1, SF3B1, EIF1AX, and SRSF2 [11]. There are some mutations detected at a low frequency (e.g., CSDM1, TTC28, TP53BP1, DLK2, and KTN1) where the clinical significance is currently unknown [12,13].

Treatment options for UM include radiotherapy (brachytherapy or teletherapy) and tumour surgery (transscleral resection, endoresection, enucleation) [14,15]. Ocular treatment is selected according to the size and location of the tumour, but there is no consensus on setting a standardised treatment protocol based on different tumour characteristics and genetic profiling.

The chorioallantoic membrane (CAM) model is considered to be a reliable in vivo experimental model for studying tumour biology and can be advantageous due to its immunodeficiency properties, allowing transplantation of various tissues without triggering an immune response or rejection of the implant. The CAM constitutes a biologically faithful model and is an increasingly valuable tool in oncological research, enabling the transplantation of various types of cancer cell lines or even human tumour tissue and providing information regarding tumour formation, angiogenesis, and metastasis. The presence of T cells can first be detected on day 11, B cells are present on day 12, and by day 18, the chicken embryos become immunocompetent [16]. Moreover, CAM allows rapid vascularisation; therefore, many cancer cell types can complete extravasation within 24 h after their implantation [17,18]. If the experiments can be completed within 17 days, ethics approval is not required. However, despite its numerous advantages, the CAM model has certain limitations. The major disadvantage is the inability of the formation of macroscopically visible colonies for some tumour cells in secondary organs due to the short span of 8–10 days between the implantation and the completion of the experiment. This predominantly affects the “slow growth” tumour cells. Another disadvantage comprises the mobility of the embryo and the smooth surface of the CAM that can sometimes lead to the motion of tumour cells and interruption of the implantation process. The latter constitutes a crucial necessity for the optimisation of the CAM model for various tumour entities [19,20].

To our knowledge, the optimisation of the chicken CAM as a reliable in vivo model for the analysis of choroidal melanoma has not yet been described in the literature. In this study, we analysed the established CAM assay using UM 92.1, UPMD2, UPMM3, and Mel270 cell lines as references and optimised the chick embryo viability in order to evaluate the tumour spheroid attachment, the implantation/growth rate, the angiogenic potential, and the changes in the tumour spheroid’s volume and metastatic capability.

## 2. Materials and Methods

Various parameters were tested to improve the viability of the chicken embryo and the growth of the implanted spheroid tumours (Table 1).

### 2.1. Cell and Spheroid Culture

UM cell lines UM92.1, UPMD2, and UPMM3 were provided by M. Zeschnigk (Institute of Human Genetics, University Hospital Essen, Germany), and Mel 270 was provided by K.G. Griewank (Department of Dermatology, University Hospital Essen, Germany). The UM cell lines were maintained in an RPMI 1640 medium supplemented with 10% foetal calf serum and 1% penicillin-streptomycin (5000 U/mL). The spheroids were generated by seeding 10 × 10^3^ or 20 × 10^3^ living cells in round-bottomed 96-well ultra-low attachment plates (Corning, Corning, NY, New York, USA) in 100 µL of the cell culture medium.

The medium was refreshed two times per week. The cell and spheroid cultures were incubated in a humidified incubator (37 °C, 5% CO_2_) for the indicated period of time. Compact spheroids could be generated on day 4 of the culture, and on day 9, the spheroids were transplanted on the CAM.

### 2.2. Chick Chorioallantoic Membrane Assay

Fertilised white chicken eggs were delivered by OVOVAC (GmbH, Burkau, Germany). In order to avoid contamination and to evaluate the effect on embryo viability, the eggs were cleaned with either distilled water or 70% ethanol. After approximately 12 h of preservation at a temperature of 21 °C, the eggs were placed in an incubator (400-RD, Bruja GmbH). The eggs were incubated in a horizontal position at a temperature of 37.8 °C and a humidity of 70% with permanent agitation (at least 3 times/day). On day four of embryonic development (ED 4), approximately 4–5 mL of albumin were removed from the apical side of the egg by positioning it vertically and using a 20-gauge needle without detaching the embryo or the yolk. This step was performed using diaphanoscopy for an easier localisation of the embryo and the yolk sac. The puncture hole was covered with Leukosilk tape. At ED 5, Leukosilk tape was applied on the upper side of the egg before an oval window of 1 × 1 cm was cut out of the eggshell to access the CAM. The window was resealed with adhesive tape.

### 2.3. Tumour Spheroid Implantation Procedure

To implant tumour spheroids, the window was reopened, and a sterile plastic ring (2 mm inner diameter) was placed into the CAM. The small plastic ring was made of various standard lab plastic pipettes (Thermo Scientific, Germany). The CAM area within the ring was gently lacerated using different techniques for the removal of the epithelium layer, including the use of a 30-gauge cannula, Nr. 11 scalpels (Aesculap, Tuttlingen, Germany), and a 25-gauge vitrectomy membrane scraper before the spheroids were inoculated. Before transplantation, the tumour spheroids were covered in different matrices and transplanted into the CAM at ED 5, 7, 9, or 11. To improve the spheroid attachment into the CAM, we compared spheroids covered in Matrigel as well as various other viscoelastic substances. Tumour spheroids covered with either Matrigel or the viscoelastic substance were placed in the middle of the ring between two large chicken vessels (Figure 1B). The window was resealed with adhesive tape, and the eggs were incubated at 37.8 °C until ED 17. Unfertilised eggs and embryos that died before ED 17 were excluded from the study. All experiments were repeated three times under the same conditions.

The Matrigel (Corning GmbH, Kaiserslautern, Germany) was thawed at 4 °C, and 80 µL of the respective cell culture medium was removed; 20 µL of it was mixed with 20 µL of matrix material and transplanted into the CAM after Matrigel polymerisation. The polymerisation of the matrix material occurred within 30 min at 37 °C.

Apart from Matrigel, various other viscoelastic substances were tested. We applied Hyalon (Johnson & Johnson Vision, Santa Ana, CA, USA), the most commonly used viscoelastic (a 1% solution of sodium hyaluronate (NaHA) with a molecular weight of 4 million daltons and a viscosity of 229,000 cP), Hyalon GV (Johnson & Johnson Vision), the modification of the Hyalon (1% NaHA with a molecular weight of 5 million daltons), Provisc (Alcon, Hattingen, Germany, 1% NaHA with a molecular weight of 1.9 million daltons), and Viscoat (Alcon, composed of 3% sodium hyaluronate with a molecular weight of 600,000 daltons).

### 2.4. Immunohistochemistry

Excised tumour spheroids were fixed in buffered formalin (Histofix). Formalin-fixed paraffin-embedded sections were cut at a thickness of 3 µm. The Melan A (clone A103, Roche, Ventana) and SOX10 (clone BC34, Biocare Medical) immunohistochemistry analysis was performed using the Ventana BenchMark ULTRA system (Ventana Medical Systems, Tucson, AZ, USA). The immunohistochemistry (IHC) analysis for Melan A was performed using the following protocol: pretreatment CC1 was performed for 36 min at 95 °C, and incubation occurred for 24 min at 27 °C. For SOX10, CC2 pretreatment was performed for 40 min at 90 °C, and incubation occurred for 32 min at 36 °C. Visualisation was conducted using the OptiView DAB System (Ventana Medical System). Thereafter, slides were scanned using a Leica DM4000B microscope with a Leica DFC290 camera and analysed using the Leica Suite version 2.8.1 (Leica Microsystems, Wetzlar, Germany). The slides were reviewed by a board-certified pathologist.

### 2.5. Fluorescence Microscopy

The vessels of the CAM were detected in ovo after the intravascular injection of 50 µL Fluorescein (Fluorescite 10% Firma Alcon) and were imaged with an Olympus BX51 microscope at 20× magnification.

### 2.6. Image Analysis

The excised tumour spheroids were imaged at ED 5, ED 11, and ED 17 using a Zeiss Primovert bright-field microscope at 40× magnification. Images were recorded using a Zeiss Axiocam 105 and ZENcore software and analysed using ImageJ Fiji (Dresden, Germany) image processing software. The sizes of the tumour spheroids and nodule explants were determined by calculating the cross-sectional area of the spheroids or explants (µm^2^).

### 2.7. Data Statistics

A statistical analysis of the data was performed using a two-way ANOVA test and Tukey’s multiple comparisons test (GraphPad Prism 8.4.3 software, GraphPad Software Inc., San Diego, CA, USA). A value of *p* < 0.05 was considered statistically significant and significance levels were indicated as * *p* < 0.05, ** *p* < 0.01, *** *p* < 0.005, **** *p* < 0.001.

## 3. Results

### 3.1. Chick Embryo Viability Pretreatments of the CAM

We compared the effect of the cleansing method using either distilled water or a 70% ethanol disinfectant on the embryo viability. Ethanol significantly reduced the viability within the first five embryonic development days (ED 5) to approximately 65% (ED 13–17) compared to 85% in the sterile-water treated group (Figure 1A).

The rich vascular system of the CAM develops within the intermediate mesodermal layer that is located between the outer chorionic epithelium and the inner allantoic epithelium. To facilitate tumour spheroid engraftment and neovascularisation, the outer epithelium must be partially removed using gentle laceration. We compared different techniques for the removal of the epithelium layer including using a 30-gauge cannula, an Nr. 11 scalpel, and a 25-gauge vitrectomy membrane scraper. The resulting vascularisation of the tumour mass was higher (from initially 20% to >80%) when a 25-gauge membrane scraper or a scalpel was used to scratch the CAM surface. Furthermore, the removal using a 25-gauge membrane scraper enabled gentler scratching, and serious injuries of the blood vessels, as well as excessive haemorrhages, could be avoided.

### 3.2. Tumour Spheroid Attachment Rate

The critical conditions for successful positioning and maintenance of the tumour spheroid within a defined area of the CAM were the use of a plastic ring and an extracellular matrix (Figure 2). The main use of the application of viscoelastic substances in ophthalmology is during cataract surgery either to prevent corneal endothelial cell loss during phacoemulsification or when implanting the intraocular artificial lens. We applied and compared Matrigel with various viscoelastic substances such as Hyalon, Hyalon GV, Provisc, and Viscoat. A medium with higher viscosity could potentially increase adhesion. The different viscoelastic substances were comparable using a spheroid tumour adhesion rate of 60%; however, the spheroid tumour attachment rate using Matrigel was as high as 90% on both ED 11 and ED 17 (Figure 2).

### 3.3. Tumour Spheroid Growth

We investigated the influence of the incubation time on the tumour attachment rate and viability of the spheroids. A transplantation with a ring plus Matrigel at ED 5 reduced the tumour attachment rate to 60%. Similarly, the viability of the chicken embryos and the volumes of the developing tumours increased when transplantation took place at a later time. Transplantation on ED 11 and resection on ED 17 were the optimal conditions. The best results concerning tumour growth were observed after the implantation of tumour spheroids formed by 20 × 10^3^ cells for all four tested cell lines. The most noticeable growth development during the experiment could be documented for the UM 92.1 spheroids and is presented in Figure 3. Furthermore, the spheroid development was compared for four cell lines: UM 92.1, Mel270, UPMD2, and UPMM3. The UM92.1 and UPMD2 cells produced compact pigmented spheroids and could be easily transported into the CAM. The UM92.1 cells produced spheroids with increasing sizes, while the UPMD2 cells aggregated into small spheroids with a constant size. After transplantation, the UPMD2 cells showed no significant increase in the spheroid size. The Mel270 cells tended to aggregate into large, flat spheroids with an increasing size but low adhesion. Spheroids from the Mel270 and UPMM3 cells revealed low pigmentation that complicated the inoculation into the CAM as well as further evaluation. The differences between the various implanted cell lines may be associated with the tumour origin.

Histological analysis of the grafted tumour spheroids at ED 17 confirmed that they were completely integrated into the CAM and had formed a nodule-like structure (Figure 4). The tumour spheroids were fully covered by the CAM layers: chorionic epithelium, vascularised mesenchyme, and allantoic epithelium (Figure 4A,B, indicated by c, v, a, respectively). We used Melan A and SOX10 immunohistochemistry tests to identify uveal melanoma cells in the nodules (Figure 4C–F). Uveal melanoma cells of the tumour spheroid spread out into the CAM (Figure 4E).

The tumour spheroid nodules showed an optimal and rapid vascularisation after transplantation on ED 11. The intravascular injection of Fluorescein revealed tumour spheroid vascularisation at day ED 17 (Figure 5). Haematoxylin staining confirmed the vascularisation of the tumour spheroid nodule (Figure 5C).

The application of Matrigel as a matrix, the use of a plastic ring for the positioning of the tumour spheroids between large vessels on the CAM, the laceration of the CAM using a 25-gauge membrane scraper, and the use of 20 × 10^³^ cells as the optimal cell number increased the overall tumour attachment rate to 80%.

## 4. Discussion

The significant similarity of the CAM to the human embryo at cellular and anatomical levels, including its rapid development, its comparatively large structures during the early developmental stages, and the accessibility of the CAM for experimental manipulation, constitute its establishment as a unique and crucial research model. There are no available literature sources for the optimisation of pre-established CAM assays for the analysis of UM tumours. Different groups report a high mortality rate of the embryos and varying success rates concerning the implantation of the tumour cells [21]. To optimise the CAM assay protocols for a reliable application in uveal melanoma research, we aimed to reduce the mortality rate of the chicken embryo [22,23].

A standardised protocol for the pretreatments of the CAM played an important role. We observed that gently scratching the CAM surface using a 25-gauge membrane scraper, putting a plastic ring on the CAM, and applying Matrigel and a cell medium to the CAM delivered the best results. Although viscoelastic substances have an advantage over other mediums due to their rheological properties, including viscosity, viscoelasticity, pseudo-plasticity, cohesiveness, and coatability, the achieved tumour mass volumes and the attachment rates were higher using Matrigel [24]. These results could be attributed to the changing viscoelastic properties of Matrigel depending on the polymer concentration and the length of time used in the experiment. It is postulated that the biophysical and viscoelastic characterisation of Matrigel varies over time, becoming more solid-like and settling to a constant state after one to three hours [25].

The external application of ethanol or other disinfectants may have a significant impact on the viability of the embryos due to the permeability of the eggshell and should be avoided. The use of distilled water in our study significantly reduced the mortality rate without any increase in contaminations. A further improvement of the survival rate of the embryos could be achieved by comparing various surgical instruments for the fenestration, thereby minimising the risk of damage during the opening of the eggs. In our study, we found that further damage could be avoided during the removal of albumen by using diaphanoscopy. Moreover, the optimisation of the tumour cell count for the spheroids to form CAM tumours may crucially affect the viability and growth potency, as shown in our results. A CAM protocol for osteosarcoma cell lines reports an improvement of the tumour take rate from 51% to 94% and an improvement in the rate of the viability of the embryos from 40% to >80% after optimisation. The results are similar to the data presented in our study, underlining the importance of every step and each procedural detail during the preparation and growth of the embryos [23].

The optimisation of the CAM assay for UM tumours and a higher rate of viability and growth of tumour masses may enable the implantation not only of tumour spheroids such as those used in our study but also of human tumour tissue for research purposes. The differences between the various cell lines tested in our study pose a challenge depending on the origin of the implanted tumour spheroids. The UM92.1 cell line showed a more robust growth development in the CAM that can likely be accredited to its emergence from a massive tumour mass that had destroyed the eye and orbit and led to extraorbital metastases [26]. In addition, the low pigmentation of the Mel270 spheroids may also be attributed to their acquisition from a large recurrent tumour after prior irradiation [26]. The role of pigmentation in primary or metastatic tumours has been discussed controversially. Studies have examined the role of melanin and its effect on cell elasticity as an important property of the cell’s efficient in vivo spread [26]. The phenomena investigated under these conditions and the in vivo nature of the CAM assay imitate and extend to the detailed molecular or biochemical behaviour of the human organism, allowing the delineation of the effects of drug administration on these organisms. The experimentation with chick embryos without authorisation from animal experimentation committees could be of interest for screening purposes or for testing not dependent on animal use.

The ability of the UM cell lines UM92.1 and Mel270 to form solid tumour spheroids and induce metastasis has already been discussed [27,28,29,30]. The adaptation of various parameters provided a reliable and reproducible in vivo model system for the analysis of uveal melanoma, including the analysis of intra-tumoural vascularisation. Notably, the efficacy of the developed protocol could be confirmed by the successful generation of solid tumours from all four UM cell lines tested in this study. The optimisation of the CAM assay with implantation of uveal melanoma spheroid tumours facilitated the reproducibility of the tested conditions. These observations confirm the capability of this in vivo model to be partially substituted for animal experiments in uveal melanoma research. The inclusion of the CAM assay into the pipelines of large-scale in vivo experiments could result in a simplified and low-cost revelation of compounds with therapeutic potential for rare tumours such as UM.

## 5. Conclusions

The chick CAM assay is an important model in cancer research. The main advantages of this assay pertain to economy and reliability. The use of this model supports a considerable replacement of animal experiments. A disadvantage is the high mortality rate of the embryos and the rejection of tumours after transplantation into the CAM. Moreover, not all tumour cells can produce macroscopically visible colonies over the short time of the experiment. However, we were able to improve the viability rate of the embryos by up to 80% by using a standardised protocol. We established strategies to raise the stability of the spheroids in the CAM, improved the vascularisation of the spheroid, and achieved a threefold increase of the transplanted spheroid volume. The optimisation of the chicken CAM in UM research may expedite preclinical tests and facilitate the development of personalised methods in oncology.

## Figures and Tables

**Figure 1 pharmaceutics-14-00013-f001:**
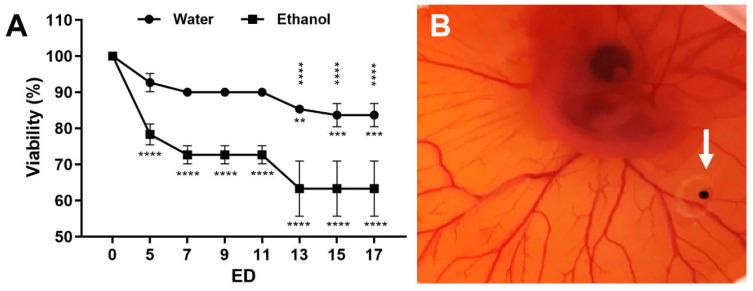
Viability of embryos. (**A**) Viability of the embryos at the indicated embryonic development day (ED) after cleaning of the eggshell with either sterile water or 70% ethanol with a mean +/− SD value of three independent experiments (each n = 20–30 eggs). The statistical significance of ED 5–17 related to ED 0 is indicated below the curves. The statistically significant differences between the water and the ethanol group are shown above the curves. (**B**) Representative in ovo image of a CAM at ED 6 with a 4× magnification. The strong vascularisation of the CAM indicates the viability of the embryo. A tumour spheroid was grafted in a plastic ring between the vessels (arrow).

**Figure 2 pharmaceutics-14-00013-f002:**
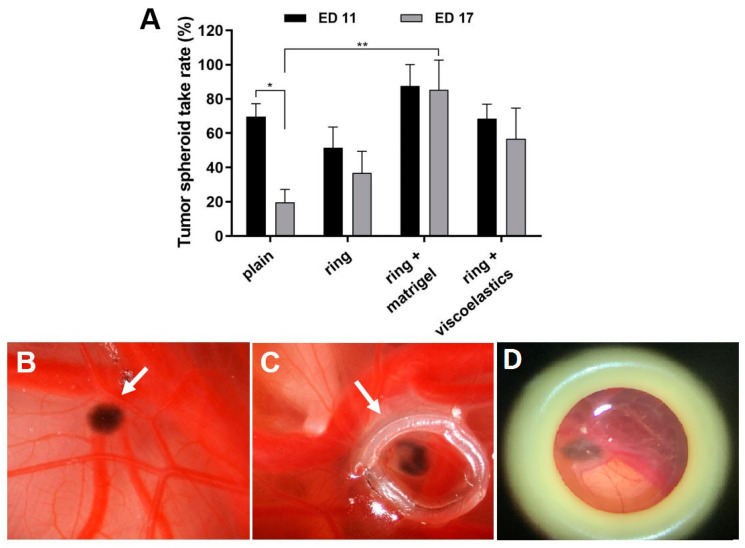
Effects of the cell matrix on tumour spheroid take rates. At ED 11, tumour spheroids were grafted into the CAM and were placed in a plastic ring (plain), or were placed in a plastic ring with either Matrigel or other viscoelastic substances (Hyalon, Hyalon GV, Provisc, Viscoat). (**A**) Analysis of the spheroid take rates at ED 11 and ED 17. Mean +/− SD from 2–3 independent experiments (each n = 12–16 tumour spheroids on the CAM). (B-D) Representative in ovo images of grafted tumour spheroids on the CAM at ED 17 (arrows); (**B**) spheroid only (plain), (**C**) spheroid in ring + Matrigel, and (**D**) spheroid in ring + viscoelastic substance Viscoat. Original 4× magnification.

**Figure 3 pharmaceutics-14-00013-f003:**
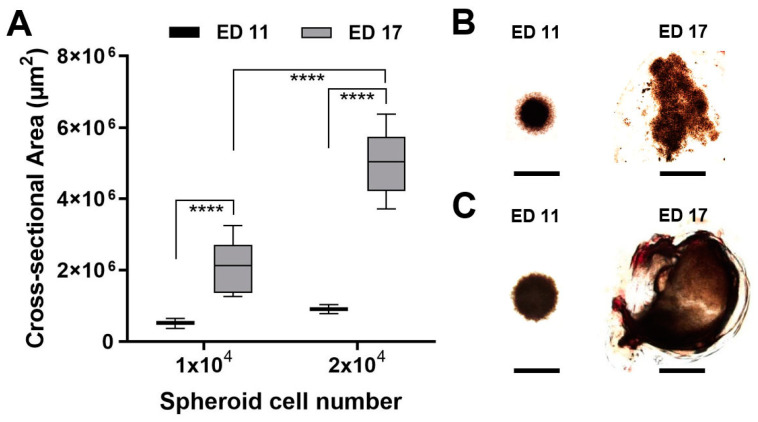
Tumour spheroid growth in the CAM assay. Tumour spheroids were grown from 1 × 10^4^ or 2 × 10^4^ UM92.1 cells and transplanted on the CAM at ED 11. The tumour spheroid nodules were explanted at ED 17 and imaged using microscopy. (**A**) The cross-sectional areas of the tumour spheroids or the explants were measured. The means +/− SD of each group with n = 10 spheroids are shown. (**B**,**C**) Representative micrographs of tumour spheroids or explants are shown. The scale bar indicates 1 mm.

**Figure 4 pharmaceutics-14-00013-f004:**
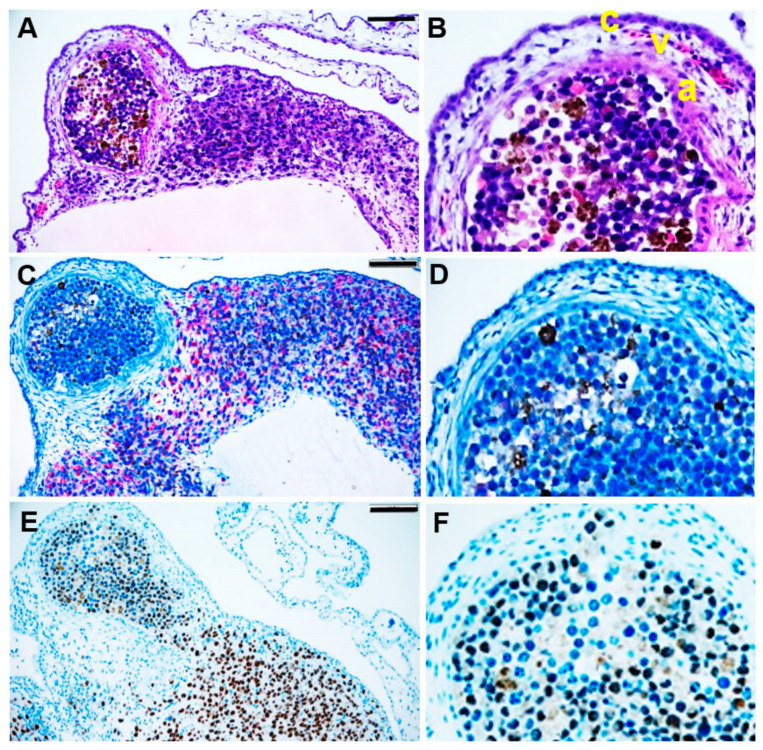
Immunohistological analysis of a tumour spheroid nodule formed in the CAM assay. At ED 17, the tumour spheroid nodule on CAM was explanted and histologically analysed. (**A**,**B**) After H&E staining, the red-brown colour indicates pigmented melanoma cells. (**C**,**D**) Immunohistological staining of human melanoma marker Melan A or (**E**,**F**) human melanocyte development marker SOX10. The red colour indicates Melan A, and the brown colour indicates SOX10 positive melanoma cells. The scale bar indicates 100 µm. (**B**,**D**,**F**) Enlarged sections of the nodules in A, C, and D, respectively. (**B**) The CAM layers: the chorionic epithelium (c), the vascularised mesenchyme (v), and the allantois epithelium (a).

**Figure 5 pharmaceutics-14-00013-f005:**
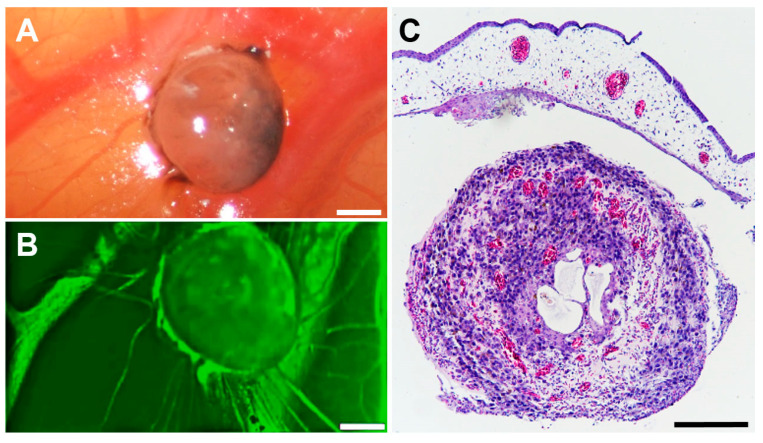
Vascularisation of a tumour spheroid nodule formed in the CAM assay. Tumour spheroid nodule on the CAM at ED 17. (**A**) Representative in ovo image of a tumour spheroid nodule on the CAM at ED 17. The scale bar indicates 100 µm. (**B**) Intravascular injection of 50 µL Fluorescein revealed vascularisation of the tumour spheroid nodule in ovo. (**C**) After H&E staining, the red colour indicates vessels. The scale bar indicates 100 µm.

**Table 1 pharmaceutics-14-00013-t001:** All tested parameters and conditions for the improvement of viability and growth of the implanted sheroid tumors.

Parameters Analysed	Conditions Tested
Number of cells for the spheroids (92.1, Mel270, UPMM3, UPMD2 cell lines)	10 × 10^3^ cells
	20 × 10^3^ cells
Day of spheroid transplantation	D5
	D7
	D8
	D11
Cleansing of the eggs	distilled water
	ethanol 70%
Fixation of spheroid	plastic ring, 2 mm
	plastic ring made from lab plastic pipets
	plastic ring and matrix
	without fixation
Laceration of the CAM for the implantation of the spheroids	30-gauge cannula
	Nr. 11 scalpel
	25-gauge vitrectomy membrane scraper
Matrix	Matrigel
	Hyalon
	Hyalon GV
	Provisc
	Viscoat

## Data Availability

Not applicable.

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
