# Peer review of "Optimisation of the Chicken Chorioallantoic Membrane Assay in Uveal Melanoma Research"

_pharmaceutics, 2021, doi:10.3390/pharmaceutics14010013_

Round 1
Reviewer 1 Report
Despite a number of papers focused on using chicken CAM and uveal melanoma cells, usually very few methodological details are given. The reviewed paper has a methodological character with the aim to optimize CAM assay for uveal melanoma research. Systematic testing of different approaches in the whole process of preparation and cultivation is of utmost importance. Authors provide clear instructions and adjustments. Suggestions are supported by sound results, some of which are rather surprising (e. g. cleansing of egg surface with distilled water instead of ethanol). Overall, the paper is clearly and logically written and provides useful methodological adjustments of the protocol not only for uveal melanoma research but any other research using CAM model.
Author Response
Thank you for your comments.
Reviewer 2 Report
The paper is very interesting and expands the knowledge about the use of chorioallantoic membrane model for tumor studies, especially for uveal melanoma.
However, some information is unclear and unlinked. In general, the results are not detailed to link with the discussion. Therefore, my main suggestion is to make a relationship between the aim paper, what is exposed and what was concluded.
Specific comments
Introduction:
1) There is a paragraph saying about molecular specification, but the paper did not explore this aspect.
2) How is the CAM model used with tumors? Even if not the UM cells, but what do you have about it in the literature? There is a lack of information about it in the Introduction.
Methodology
Maybe, it remained to highlight which parameters were analyzed to improve the viability of the model. It looks like everything has changed, but it's not clear that it was changed to see which would have the greatest viability. Perhaps a table could help to better understand all the parameters that were varied (suggestion).
3) the last phrase of section 2.2 is not clear. What means?
Results
As the title is "optimization of the CAM assay", any optimization would be good to be explored. You changed the method of scratch the membrane but did not provide information about the rate of improvement, the description of effects, or more detailed results.
4) Figure 2 - It is not clear how did you measure the take rate.
5) line 237 - I think that you would like to say Figure 4 and not Figure 2.
Discussion
6) It remained to explore a little about: how much better are your results when compared to each other? And how much better are they compared to literature?
7)You are not exploring in-depth the differences among the cell lines in the results. However, there is a great discussion about them in the "discussion" section. Maybe, it will be good to link each other.
I am sure that if you explore more your data, the paper will be very improved.
Author Response
Thank you very much for your valuable comments. All reviosions were made and are marked with yellow in the text.
Introduction:
1) There is a paragraph saying about molecular specification, but the paper did not explore this aspect.
Answer: Thank you for your comment. The paper focuses in establishing a standardized protocol for optimizing the viability and growth of CAM assays for uveal melanoma cells and spheroids. The molecular and further behavioral characteristics of uveal melanoma tumors were not studied in this experiment and therefore not reported in the paper. The CAM assay may simplify the exploration of molecular and biochemical traits of various tumor types in the future.
2) How is the CAM model used with tumors? Even if not the UM cells, but what do you have about it in the literature? There is a lack of information about it in the Introduction.
Answer: Thank you for your comment. The main advantages and disadvantages of the CAM as an experimental model in cancer research are analysed in the discussion. The growth characteristics and cellular behavior are reported in the introduction. Line 69-71 were added: The CAM constitutes a biologically faithful model and an increasingly valuable tool in oncological research, enabling the transplantation of various types of cancer cell lines or even human tumor tissue and providing information regarding tumor formation, angiogenesis and metastasis.
Methodology
Maybe, it remained to highlight which parameters were analyzed to improve the viability of the model. It looks like everything has changed, but it's not clear that it was changed to see which would have the greatest viability. Perhaps a table could help to better understand all the parameters that were varied (suggestion).
Answer: Thank you for your comment. Table was added. Line 92.
3) the last phrase of section 2.2 is not clear. What means?
Answer: Thank you for your comment. The last sentence was deleted to avoid confusion
Results
As the title is "optimization of the CAM assay", any optimization would be good to be explored. You changed the method of scratch the membrane but did not provide information about the rate of improvement, the description of effects, or more detailed results.
Answer: Thank you for your comment. Regarding the results of the different laceration techniques, we would like to refer you to section 3.1 and specifically to lines 194-198: “The resulting vascularisation of the tumour mass were higher (from initially 20% to >80%), when a 25 gauge membrane scraper or a scalpel was used to scratch the CAM surface. Furthermore, the removal with the 25 gauge membrane scraper enabled a more gentle scratching and serious injuries of the blood vessels as well as excessive haemorrhages could be avoided.”
4) Figure 2 - It is not clear how did you measure the take rate.
Answer: Thank you for your comment. As mentioned in the figure legend, the presented data are mean values with standard deviation from three experiments. Each experiment was conducted after transplantation of approximately 12-16 tumor spheroids. Therefore, the mean values represent the attachment of the 36-48 (12-16x3) replicates for each tested condition.
5) line 237 - I think that you would like to say Figure 4 and not Figure 2.
Answer: Thank you for your comment. Line 245: Figure 2 was changed to Figure 4
Discussion
6) It remained to explore a little about: how much better are your results when compared to each other? And how much better are they compared to literature?
Answer: Thank you for your comment. The results between the different preparation techniques were analyzed in the text as well as in the graphics and images. A comparison in the literature to CAM protocols for other tumor cell types was added, although potential differences could be attributed to the behavioral characteristics of the different tested tumor entities. Line 296-299 was added: A CAM protocol for osteosarcoma cell lines reports an improvement of tumor take rate from 51% to 94% and of viability of the embryos from initially 40% to >80% after optimization. The results are similar with the presented data of our study, underlying the importance of every step and procedural details during preparation and growth of the embryos.
7)You are not exploring in-depth the differences among the cell lines in the results. However, there is a great discussion about them in the "discussion" section. Maybe, it will be good to link each other.
I am sure that if you explore more your data, the paper will be very improved.
Answer: Thank you for your comment. We tried to avoid adding information about the characteristics of the tumor cell lines in the results section and only presented the relevant data. As you mentioned the differences between the various tested cell lines were analyzed in the discussion. Line 233-234 was added to link the results to the discussion accordingly: The differences between the various implanted cell lines may be associated with the tumor origin.
Reviewer 3 Report
The authors describe the improvement of Cam assay for the characterization of cancerogenic potential of uveal melanoma cells. The paper takes up an already published and widely used protocol without adding any new evidence. In the introduction, the Authors declare “To our knowledge, the optimization of the chicken CAM as a reliable in vivo model for the analysis of choroidal melanoma has not yet been described in the literature “ while the discussion section refers 3 different papers about UM cell line “The ability of the UM cell line UM92.1 and Mel270 to form solid tumours and induce metastasis into 304 the CAM has already been discussed [27, 28, 29, 30]. “
Thus the authors should report this paper in the introduction and analyze in detail the advantages of their protocol.
In results section, no data are shown related to the different cell lines and the similarity or differences of their behaviors in CAM assay in terms of growth, apoptosis, engraftment, histology.
Does Fig 1 B show the engraft of tumor cells on CAM? From image 1B the embryo cam seems just hinted at and in any case above the point of engraft. Do authors comment on this image?
This referee does not think that this paper is suitable for publication.
